# Perturbation of Cortical Excitability in a Conditional Model of PCDH19 Disorder

**DOI:** 10.3390/cells11121939

**Published:** 2022-06-16

**Authors:** Didi Lamers, Silvia Landi, Roberta Mezzena, Laura Baroncelli, Vinoshene Pillai, Federica Cruciani, Sara Migliarini, Sara Mazzoleni, Massimo Pasqualetti, Maria Passafaro, Silvia Bassani, Gian Michele Ratto

**Affiliations:** 1National Enterprise for NanoScience and NanoTchnology (NEST), Istituto Nanoscienze, Consiglio Nazionale delle Ricerche (CNR) and Scuola Normale Superiore Pisa, 56127 Pisa, Italy; didilamers@gmail.com (D.L.); silvia.landi@in.cnr.it (S.L.); roberta.mezzena@sns.it (R.M.); vinoshene.pillai@sns.it (V.P.); fedz7987@gmail.com (F.C.); 2Istituto di Neuroscienze, Consiglio Nazionale delle Ricerche (CNR), 56124 Pisa, Italy; laura.baroncelli@in.cnr.it; 3Department of Developmental Neuroscience, IRCCS Stella Maris Foundation, 56128 Pisa, Italy; 4Unit of Cellular and Developmental Biology, Department of Biology, University of Pisa, 56127 Pisa, Italy; sara.migliarini@unipi.it (S.M.); massimo.pasqualetti@unipi.it (M.P.); 5Institute of Neuroscience, CNR, 20854 Vedano al Lambro, Italy; sara.mazzoleni2@unimi.it (S.M.); maria.passafaro@in.cnr.it (M.P.); silvia.bassani@in.cnr.it (S.B.); 6NeuroMI Milan Center for Neuroscience, University of Milano-Bicocca, 20126 Milano, Italy

**Keywords:** PCDH19 epilepsy, girls clustering epilepsy, autism spectrum disorder, slow wave activity, excitation: inhibition ratio, NREM sleep, epilepsy

## Abstract

PCDH19 epilepsy (DEE9) is an X-linked syndrome associated with cognitive and behavioral disturbances. Since heterozygous females are affected, while mutant males are spared, it is likely that DEE9 pathogenesis is related to disturbed cell-to-cell communication associated with mosaicism. However, the effects of mosaic PCDH19 expression on cortical networks are unknown. We mimicked the pathology of DEE9 by introducing a patch of mosaic protein expression in one hemisphere of the cortex of conditional PCDH19 knockout mice one day after birth. In the contralateral area, PCDH19 expression was unaffected, thus providing an internal control. In this model, we characterized the physiology of the disrupted network using local field recordings and two photon Ca^2+^ imaging in urethane anesthetized mice. We found transient episodes of hyperexcitability in the form of brief hypersynchronous spikes or bursts of field potential oscillations in the 9–25 Hz range. Furthermore, we observed a strong disruption of slow wave activity, a crucial component of NREM sleep. This phenotype was present also when PCDH19 loss occurred in adult mice, demonstrating that PCDH19 exerts a function on cortical circuitry outside of early development. Our results indicate that a focal mosaic mutation of PCDH19 disrupts cortical networks and broaden our understanding of DEE9.

## 1. Introduction

PCDH19 epilepsy (Developmental and Epileptic Encephalopathy 9, DEE9 OMIM # 300088) is an epileptic encephalopathy likely to be the second most frequent genetic cause of epilepsy in females [1], and it is associated with intellectual disability, autism spectrum disorder (ASD) [2,3,4] and late-onset schizophrenia [5]. Seizures originate early in infancy and can be focal or generalized, usually occurring in clusters [6,7]. Epilepsy ameliorates with adolescence [6], while behavioral disturbances become the most prominent feature of the disease [4].

The syndrome is caused by mutations in the X-linked gene Protocadherin 19 (Pcdh19) [8]. The pattern of inheritance of DEE9 is highly unusual: unlike in most X-linked disorders, heterozygous females are severely affected while hemizygous males are spared [8]. Heterozygous females, due to random X inactivation, express the protein in a mosaic and this mosaicism underlies the phenotype of the disease. Indeed, males with somatic mutations—unlike fully mutant males—have been shown to display a phenotype [3,9,10].

Protocadherins have been implicated in circuit formation and maintenance, cell-cell adhesion, synaptic connectivity, plasticity and cell signaling. PCDH19 is mainly expressed in the hippocampus and layers II/III and V of the cortex [8,11,12,13]. Since PCDH19 is present in synapses [11,12,13] and regulates GABAA related inhibition [14,15], it is likely important for synapse formation and function. A cellular interference hypothesis has been proposed [3], whereby the co-existence of wild-type and knockout neurons interferes with normal cell-to-cell communication, possibly because the interaction between PCDH19 and N-cadherin disrupts synaptic targeting, leading to impaired synaptic plasticity [13].

Mouse models of DEE9 have been characterized, and these studies have offered support for the notion that PCDH19 regulates cell adhesion and network formation. Although no gross morphological abnormalities were found, the brain displayed a peculiar pattern with PCDH19 positive and negative neurons present in separate patches [11,12]. An important recent study outlined the presence of a considerable impairment of hippocampal circuitry [13], suggesting PCDH19 as a regulator of network formation and maintenance. At present, it is unknown whether cortical network wiring and computation are also affected by mosaic PCDH19 expression. Here, we aim to answer this question using in vivo electrophysiological and imaging studies in a novel mouse model of the disease. Local field potential (LFP) recordings in anesthetized animals demonstrated transient episodes of hyperexcitability and disrupted slow wave activity (SWA) in mosaic PCDH19 patches in the brain. Importantly, these phenotypes were observed even if the mosaicism was introduced in adult mice, suggesting that the gene is also important for network maintenance and homeostasis. Finally, we demonstrated that network activity in mosaic tissue is shaped by low synaptic coupling, an increased excitatory to inhibitory ratio and an increased variability of neuronal activity with a sizable population of hyperactive neurons that might trigger episodes of epileptiform activity. 

## 2. Materials and Methods

### 2.1. Animals

Mice were housed in a 12 h light:dark cycle (light on at 7 a.m. and light off at 7 p.m.) with ad libitum access to food and water. All experiments were performed between 8:30 a.m. and 6 p.m. All procedures were approved by the Italian Ministry of Health (Permit number 465-2018-PR).

### 2.2. Viral Injections

Mosaic PCDH19 expression was induced in the occipital cortex of PCDH19 conditional knockout mice by focal injection of an AAV expressing EGFP-Cre recombinase (Addgene #105545-AAV1; pAAV.CMV.HI.eGFP-Cre.WPRE.SV40). Control mice were either non-floxed littermates injected with EGFP-Cre recombinase or floxed littermates injected with an AAV expressing EGFP (Addgene #105530-AAV5; pAAV.CMV.PI.EGFP.WPRE.bGH). Mice were injected either directly after birth (P1-3) or as adults for the experimental group ‘Mosaic adults’ (>P60). Mice employed in the Ca^2+^ imaging experiments were injected twice: they received the Cre recombinase AAV vector at P1 and a second injection at about P40 with a genetically encoded calcium sensor (Addgene #100854-AAV1; pAAV.Syn.NES-jRGECO1a.WPRE.SV40). See Appendix A for details.

### 2.3. In Vivo Local Field Potential Recordings

Mice were injected intraperitoneally with urethane (dissolved in 0.9% NaCl; 0.08 mL per 10 g mouse). Throughout the procedure, the state of the animal was regularly assessed, and an additional dose of urethane was administered if needed (10% of the initial dose). Recordings lasted around 2–4 h and at the end of the experiment deeply anesthetized mice were sacrificed by cervical dislocation without regaining consciousness. 

### 2.4. EEG Recordings in Behaving Mice

Mice were implanted with an EEG headmount (containing 2 EEG channels and 1 EMG; #8201; Pinnacle Technology, Lawrence, KS, USA) one week prior to the recording. Mice were habituated to the recording room for at least 24 h before being recorded for at least 24 h. A 100× pre-amplifier (#8202-SL; Pinnacle Technology) was connected to the headmount. The signals were routed to a data acquisition system (#8206, Pinnacle Technology) which provided additional filtering (details in Appendix A).

### 2.5. In Vivo 2-Photon Calcium Imaging

Mice prepared for the imaging experiments were implanted with a perforated glass window on the right hemisphere. The LFP was recorded by a glass electrode inserted through the coverslip perforation. LFP traces were sampled at 2 kHz. Imaging was performed with a 2-photon microscope (Ultima IV; Bruker, Billerica, MA, USA) with a 20–30 mW power on the sample.

### 2.6. Data Analyses

Electrophysiology and imaging data were analyzed using custom MATLAB code available on GitHub (https://github.com/DidiLamers/PCDH19_ZebraExplore (accessed on 18 May 2022) and https://github.com/DidiLamers/PCDH19_CalciumImaging (accessed on 18 May 2022). Further details in Appendix A.

## 3. Results

### 3.1. Behavioural Phenotype of PCDH19 Mice

We created a patch of mosaic PCDH19 expression by injecting an AAV expressing EGFP-Cre recombinase into the right occipital cortex of a conditional mouse where exon 3 of the murine gene is flanked by a pair of LoxP sequences (Figure 1A and Appendix A). The opposite hemisphere was not injected and acted as an internal control. The control group was obtained by injecting an EGFP AAV in floxed littermates or an EGFP-Cre AAV in non-floxed littermates. This procedure created two coexistent types of mosaicism: a large patch of brain tissue around the injection site with a reduced PCDH19 expression surrounded by tissue with normal PCDH19 expression and, within this patch, a mosaic of cells transduced by the virus that are likely PCDH19 negative and wild-type cells that have not been transduced by the virus. We confirmed the reduction of PCDH19 by quantitative Western Blot and in situ hybridization (Appendix A). 

Mosaic mice injected at P1 showed a significant number of unexpected deaths compared to that of their control littermates (Figure 1B; *p* < 0.01) between P20 and P40, the period corresponding to early and middle adolescence in mice [16]. We tested the behavior of PCDH19 mosaic mice (P60–90) and age-matched controls by automated video recordings in an open field arena. Mice were placed in the arena at midday and, after 24 h of habituation, they were monitored for a period of several days in a quiet environment. PCDH19 mosaic mice showed a hyperactive behavior characterized by enhanced locomotion at night and reduced sleep (Figure 1C–F). 

### 3.2. Hyperexcitability of PCDH19 Mosaic Mice

We recorded local field potentials (LFP) from layers II/III of the occipital cortex in anesthetized mice at the center of the patch of EGFP-Cre fluorescence and in the symmetric contralateral site where PCDH19 expression is normal. The second recording site provides an internal control for each recorded mouse. The most prominent feature of LFP was slow wave activity (SWA) in the δ (0.5–4 Hz) frequency band, a hallmark of deep sleep in mice and humans preserved under urethane anesthesia [17,18]. 

SWA consists of ‘up states’ (US), characterized by neuronal depolarization and high firing activity, and ‘down states’ (DS) characterized by network silence [17,18].

Over imposed on this pattern, we observed signs of a hyperexcitable phenotype in the form of sustained oscillations in the β (9–25 Hz) frequency band (Figure 2A), while in controls, hardly any firing occurred during DSs [17,18]. Interestingly, epileptiform bursts of similar morphology were observed in a model of focal cortical dysplasia caused by the localized ablation of the gene PTEN [19]. These events occurred more frequently than three bursts per hour in mice injected at P1 and recorded at around P25, (five out of seven mice, Figure 2A) and in mice recorded at P60 (5 out of 11 mice). β bursts were extremely rare in controls as only 3 out of 17 mice exhibited these features at a frequency lower than two bursts per hour. In contrast, in PCDH19 mosaics occurred at a frequency of 3 to 23 episodes per hour. This activity was observed in both hemispheres (Figure 2A), thus suggesting interhemispheric propagation of the hyperactive bursts beyond the area carrying the mutation (Figure 2B,C). Importantly, mice injected as adults showed a similar phenotype (four out of nine exhibited more than three β bursts per hour).

A second form of hyperexcitability is manifested by large amplitude peaks reminiscent of interictal spikes [20,21] (Figure 2B, left). Their amplitude and brief duration are indicative of hypersynchronous neuronal activity. Such events were observed in five out of seven mice recorded at P25 and in 5 out of 11 mice recorded at P60. The frequency of this activity ranged from 53 to 323 peaks per hour in the injected hemisphere (Figure 2B; right), peaks had an average duration of 0.23 ± 0.004 s and an amplitude of 0.49 ± 0.09 mV. Of the four mice recorded at P60 that displayed peaks, three also displayed β oscillations. Interestingly, 81 ± 8% of all the observed peaks occurred while the control hemisphere was in a US (significantly more than chance, one sample t test *p* < 0.05, see Figure 2B left). Such a phase locking of hypersynchronous spikes to USs has also been shown for interictal spikes evoked by the GABAA competitive antagonist bicuculline [22]. Hypersynchronous peaks also occurred in mice injected as adults (Figure 2B; right), thus suggesting that PE may not merely be a developmental disease and that the gene also has a function later in life.

The β-band bursts also occurred in behaving mice, as shown by chronic EEG recordings performed for at least 24 h. Transient oscillations in the β band were frequently observed in mice that were injected at P1 and recorded as young adults (Figure 3). Of the 11 recorded mice, 3 showed frequent bursts. Interestingly, 87% (significantly more than chance, one sample t test *p* < 0.01) of the β-oscillations in the injected hemisphere of mosaic animals occurred during NREM sleep, suggesting a coupling of this activity to SWA. 

### 3.3. PCDH19 Mosaicism Causes Disruption of SWA and Network Synchronization

SWA is disrupted in the mosaic hemisphere (Figure 4A), as quantified by the root mean square (RMS) of the LFP signal filtered in the δ (0.5–4 Hz) range (Figure 4B). All recorded mice had reduced δ band activity in the injected hemisphere compared to the control hemisphere (Figure 4C). Male and female mosaic mice showed a similar phenotype and therefore data were grouped together (Appendix A). Other metrics of SWA were also significantly reduced (Appendix A). 

Interestingly, mice injected as adults also exhibited reduced SWA (Figure 4C and Appendix A). This is in line with our previous findings of a hyperexcitable phenotype when mosaic expression is induced after the plastic period. The phenotype of the mice recorded in adolescence at P25 is milder since the SWA reduction was not significant (Figure 4C). 

A reduction of synaptic strength and overall connectivity of the network underlies the decrement in slow wave strength occurring naturally over the course of sleep [23,24]. Such a reduction in functional connectivity is associated with a reduction of the slope of the transitions to and from each US. This change is due to the reductions of the rate of recruitment and de-recruitment of neurons to and from the US. To test whether a reduced connectivity might also underlie the SWA reduction in PCDH19 mosaic mice, we calculated the US slope (Figure 5A, [24]). In all experimental groups, slopes were significantly reduced in the PCDH19 mosaic. Figure 5 depicts the maximum slope of the second segment of up states, but results were similar for the maximum slope of the first segment and for the average slopes (Appendix A). The differences were large in every mouse injected at P1 and were significant also in mice injected in adulthood (Figure 5B). Strikingly, the mice recorded at adolescence hardly presented a phenotype (Figure 5B), just as they did not for the reduction in SWA. Our data suggest that the reduction of SWA and of US slope are tightly coupled, as previously reported [23,24], and these are signs of reduced network coupling during SWA.

Such a reduction in synaptic strength might lead to lower neuronal activity and firing. With this in mind, we extracted unit activity from our LFP recordings (Figure 6A). We found a significant reduction in unit frequency in mice recorded in adulthood after being injected immediately after birth (Mosaic pups) or at P30 (Mosaic adults, Figure 6B). Interestingly, we also found that the percentage of units occurring within USs was significantly reduced in these mice (Figure 6C). This demonstrates a lower synchronization of the network to the δ cycle, consistent with lower functional connectivity in the adult PCDH19 mosaics during deep sleep. Note however how units recorded in mice that displayed frequent β bursts were better synchronized to USs compared to other mosaic animals (black filled dots in Figure 6C). 

### 3.4. PCDH19 Mosaic Animals Have an Increased Excitation to Inhibition Ratio

Both epilepsies and ASDs have been associated with a relative increase of excitation to inhibition [25,26] and a recent study has demonstrated a correlation between E:I ratio and the slope of the power spectrum of LFP recordings below 50 Hz [27], with a steeper slope resulting from relatively more inhibition. Indeed, PCDH19 mosaic animals displayed a significant reduction in the exponent of the fit exponential function (Appendix A and Figure 7, Appendix A). The differences were also significant for mice recorded at P25 but not for mice injected in adulthood. This suggests a slight increase in E:I ratio in our mice that is already present in adolescence. Indeed, when we applied γ-aminobutyric acid (GABA) to our P25 mice at the end of the experiment, the exponent dramatically increased in each recorded animal (Figure 7, inset; and Appendix A). This implies that, as expected, increasing tonic inhibition by GABA application reduces the E:I ratio. Interestingly, GABA application also increased SWA in terms of the RMS signal strength in the 0.5–4 Hz range (Appendix A; see also the example LFP trace in Appendix A, right panel). 

### 3.5. In Vivo 2-Photon Calcium Imaging

We employed in vivo 2-photon calcium imaging to quantify the activity of single neurons. As in the previous experiments, mice were injected with an AAV carrying EGFP-Cre recombinase directly after birth and recorded under urethane anesthesia around P60. Two weeks prior to the experiment, mice were injected with an AAV expressing the red sensor jRGECO1a [28]. A representative field of view is shown in Figure 8A, with ‘red neurons’ (wild-type; expressing only jRGECO1a) and ‘green neurons’ (knockout; expressing jRGECO1a as well as nuclear EGFP-Cre recombinase). We imaged calcium transients and recorded the LFP in the same hemisphere to evaluate the synchronization of calcium transients to USs. In control mice, the fluorescence fluctuations averaged over the entire field of view were closely phase locked to SWA (Figure 8B). Individual neurons displayed transients more rarely than the US transitions (Figure 8C), since firing occurs only sparsely with most neurons being silent or producing only one action potential during each US [29]. The rarity of the transients observed in controls (median frequency under 0.02 Hz, Figure 8C) suggests that, in our conditions, we can only detect calcium transients in correspondence of multiple spikes. Therefore, transient detection is intrinsically biased toward the population of more active cells. The distribution of transient frequencies has a larger dispersion in PCDH19 mosaics compared to controls (Figure 8C; variance Bartlett’s test: *p* < 0.0001). Interestingly, a similarly increased variability of neuronal activity has been observed in a mouse model of Alzheimer’s disease [30]. There, the large variability was accompanied by an increase in hyperactive neurons (neurons with > 4 calcium transients per minute). This population of hyperactive neurons might trigger seizures associated with Alzheimer’s disease and possibly a similar phenomenon is taking place in PCDH19 mosaic mice. Indeed, in PCDH19 mosaic mice, 26.1% of neurons were hyperactive, while only 7.8% of cells were hyperactive in the control (dotted line in Figure 8C). The hyperactive neurons might underlie the observed hyperactive bursts in the mosaic tissue. 

Finally, in agreement with the SU analysis (Figure 6), calcium transients in mosaic animals were less synchronized to USs than in controls (Figure 8D). Interestingly, mosaic animals with a hyperexcitable phenotype (>5 β-oscillations per hour) seemed to have a higher degree of synchronization with SWA than non-epileptic mosaics (Figure 8D; magenta versus blue dots). This difference is statistically significant (*p* < 0.0005), and it confirms the results from our LFP recordings. 

## 4. Discussion

Studies in animal models show that PCDH19 expression is organized in relatively large patches [11,12] and that the loss of PCDH19 affects neuronal migration [31]. Given this background, it is not surprising that some DEE9 patients exhibit cortical dysplasia and folding abnormalities [11,32,33,34]. Thus, it is likely that mosaicism is not only represented by a random mix of neurons with the two genotypes, but also by mosaicism on a large scale with patches of reduced PCDH19 expression interspersed in brain tissue with normal PCDH19 expression.

Interestingly, we observed that, even if PCDH19 mosaicism occurs only in a relatively small portion of the cortex, these mice show a hyperactive behavior and a complex electrophysiological phenotype, consistent with the clinical features of the disease [35].

Hyperactivity and epilepsy are frequent co-morbidities [35,36] and, indeed, we observed two signs of hyperexcitability in our mouse model: oscillations in the β (9–25 Hz) band and large amplitude ‘hypersynchronous’ peaks. We never recorded true ictal events, which given the short duration of our recordings (2–4 h) is not surprising. The phenotype we observed in adult mice is relatively mild, consistent with the observation that in patients, epilepsy fades in severity in adulthood [6]. Interestingly, the mortality peak of our mosaic mice was around adolescence and at this time mice exhibited a more severe electrophysiological phenotype.

Our data show a clear relationship between SWA and epileptiform activity: hypersynchronous peaks tended to occur during USs, and β-bursts, recorded in the EEGs, occurred mainly during NREM sleep. Indeed, it is known that seizures [37] and interictal spikes [38] are more likely to occur during slow wave sleep than during REM sleep or wake and that they are associated with slow waves [39,40]. In fact, in DEE9 patients, seizures occur primarily during sleep [41]. Altogether, these data confirm that the mouse model recapitulates important features of patients and that it provides a model to study how hyperexcitable activity arises in PCDH19 mosaic tissue and how it can be mitigated by pharmacological treatments.

In patients, the phenotype changes in adolescence, when seizures tend to decrease in frequency and the behavioral features become the most prominent aspect of the disease [4,6]. This developmental switch is also present in our data, since epileptiform bursts are more frequent in adolescents (P25) than in adults (P60), while the disruption of SWA observed in adults is hardly present in adolescent mice. We could speculate that the reduced synaptic strength and synchronicity during SWA could represent a compensatory mechanism to prevent hypersynchronous activity triggered by the transitions to USs. This scenario is compatible with the observation that mice displaying the highest degree of synchronization with SWA were also characterized by frequent β-oscillations.

The progressive loss of SWA in mosaic mice possibly reflects a different function of PCDH19 in the developing network versus a mature network. Our data support the idea that PCDH19 plays a role in synapse and circuit maintenance throughout life, since mice injected with Cre-AAV as adults displayed a phenotype, both in terms of hyperexcitability and SWA disruption. This data is consistent with the recent finding that reduced expression of the clock gene BMAL1 in adult mice causes a loss of PCDH19 expression and an increase in susceptibility to epilepsy [42]. These findings have important implications for patients since some of the symptoms are caused by the role of PCDH19 in the mature brain and could possibly be reversed by correcting gene function later in life. 

SWA aids memory consolidation during sleep [43,44] and is important for cognitive functions since it has been hypothesized to restore overall synaptic strength of the network [45] by operating synaptic downscaling [46,47,48]. High-firing neurons reduce their activity during NREM sleep, while neurons that fire at a low rate increase theirs. Thus, sleep homogenizes the firing rate distribution [49]. We observed a large variability of neuronal activity in PCDH19 mosaic tissue. This might be partly caused by the lack of slow wave sleep’s homogenizing effect on firing rates, leading to an unstable network. Disruptions of SWA in PCDH19 patients would therefore likely affect their cognitive performance and might partly underlie the intellectual disability frequently associated with the disease. 

Growing evidence suggest that sleep is affected in PCDH19 patients as they frequently experience difficulties in falling and staying asleep [50]. Recently, a polysomnography study on a PCDH19 patient complaining of prolonged nocturnal awakenings, revealed abnormal NREM sleep episodes. SWA co-occurred with rapid eye movements, a rare state dissociation where NREM sleep appears to be intruded by REM sleep [51]. Our behavioral assessment of sleep and the electrophysiological assessment of SWA suggest that sleep disturbances are also a key component of the PCDH19 phenotype in the mouse model. 

The analysis of activity of PCDH19 mosaic has shown an increased E:I ratio as well as reductions in SWA, up state slope and synchronicity of neuronal firing to SWA. Many of the observed effects are common features of ASDs, epilepsy, and schizophrenia [25,52,53]. Interestingly, schizophrenia is a late-onset feature of DEE9 [53]. Obviously, a proper E:I ratio is crucial for network computation [25] and elevated E:I ratio has been demonstrated to impair neuronal information processing in mice by saturating the input–output curve [53]. Additionally, a proper E:I balance is needed to maintain stable firing rates in a feedforward neural network [54]. 

The reduced synaptic coupling of the PCDH19 network appears to be in contrast with the hyperexcitable phenotype. This apparent contradiction has been observed elsewhere: for example, in Rett syndrome firing rates appear to be reduced [55]. Additionally, increased inhibitory activity has been linked to epileptiform activity [56,57]. Indeed, the relationship between synaptic strength and firing rates is not clear-cut. Reductions in synaptic coupling do not necessarily reduce firing rates, nor does an increased E:I ratio inevitably increase firing rates [27]. In fact, many ASD mouse models show no changes or even reductions in pyramidal firing rates, despite reduced inhibition and susceptibility to seizures [26]. It is conceivable that network homeostasis resulting from an increased E:I ratio cannot maintain stable firing rates [26,58]. Such a network would display a large variability in activity, as is the case in PCDH19 mosaic mice, and consequently be unstable. Much evidence suggests a link between epileptic-like activity and increased variability of activity driven by a population of hyperactive neurons. In Alzheimer’s disease [30,59], a population of hyperactive neurons not only displays a high firing rate but also highly correlated firing, increasing the risk of seizure-like activity [30].

Here we presented an extensive physiological characterization of cortical activity in a mouse model of PCDH19 epilepsy. These mice display behavioral hyperactivity, signs of hyperexcitability and disrupted SWA. We additionally found that DEE9 is not merely a neurodevelopmental disease and that some symptoms might arise from disturbed PCDH19 function in the mature network. Finally, our data suggest that the PCDH19 mosaic network is less strongly coupled, less synchronized, displays increases in E:I ratio and has a large variability of neuronal activity with a population of hyperactive neurons. We propose that the behavioral and cognitive problems derive from a reduced network coupling, while the bursting of hyperactive neurons underlies the hyperexcitable phenotype. Several aspects of the electrophysiological signatures that we have found, such as the presence of isolated interictal spikes and β-bursts and the amplitude and stability of slow wave oscillation, could potentially be used as biomarkers of the disease and could represent useful checkpoints for the evaluation of the efficacy of pharmacological treatments.

## 5. Conclusions

Pcdh19 is becoming one of the most clinically relevant genes in epilepsy, yet little is known about its function. Here, we performed a thorough electrophysiological characterization of cortical activity in a novel focal mouse model of the disease and found that a mosaic network not only displays signs of hyperexcitability, but also severe disruptions of SWA, a crucial component of NREM sleep. Additional analyses suggested an increased E:I ratio and reduced coupling of the network. Furthermore, we have shown that PCDH19 is also required during adult life, as its loss in early adulthood caused the onset of the disease. Finally, calcium imaging experiments implied that the hyperexcitable phenotype might derive from a population of hyperactive neurons present in the mosaic network. 

## Figures and Tables

**Figure 1 cells-11-01939-f001:**
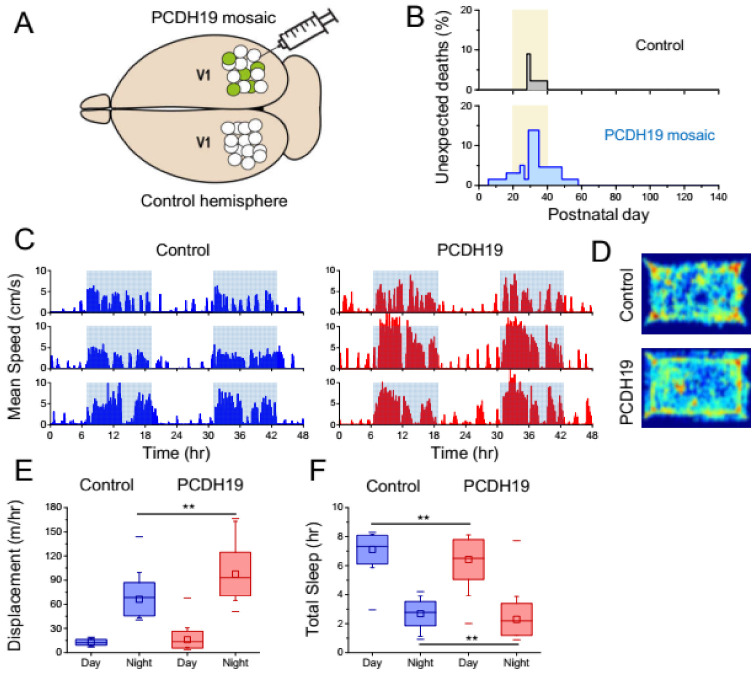
(**A**) A focal mosaic of PCDH19 expression was created by injecting an AAV expressing EGFP-Cre recombinase into the right visual cortex (V1) of the PCDH19flox/flox mouse. The opposite, non-injected hemisphere was used as the internal control. (**B**) PCDH19 mosaic mice experienced a transient period of unexpected mortality. Data collected from 70 mice injected at P1 (blue) differs significantly from that of control littermates (N = 48 mice, black; Mantel-Cox test *p* < 0.01). Yellow area indicates the period of adolescence in mice. (**C**) PCDH19 mice have a normal diurnal alternation of resting phases during the day (light on) and of high locomotor activity during the night (light off, shaded area; darkness from 7 PM till 7 AM). Bars represent the speed averaged in 5 min bins measured in three control and three PCDH19 mice in a 48-h period. Recordings start at 12 AM. (**D**) Density maps showing the arena occupation during a 2-h period of the active phase. (**E**) PCDH19 mice are hyperactive during the night phase. (**F**) PCDH19 mice spend less time than controls in resting states that can be ascribed to sleep (N = 8 for both control and PCDH19 mice; data pooled from 36 and 38 days of video recording). Significant differences are indicated with asterisks (** *p* < 0.02, Mann–Whitney test).

**Figure 2 cells-11-01939-f002:**
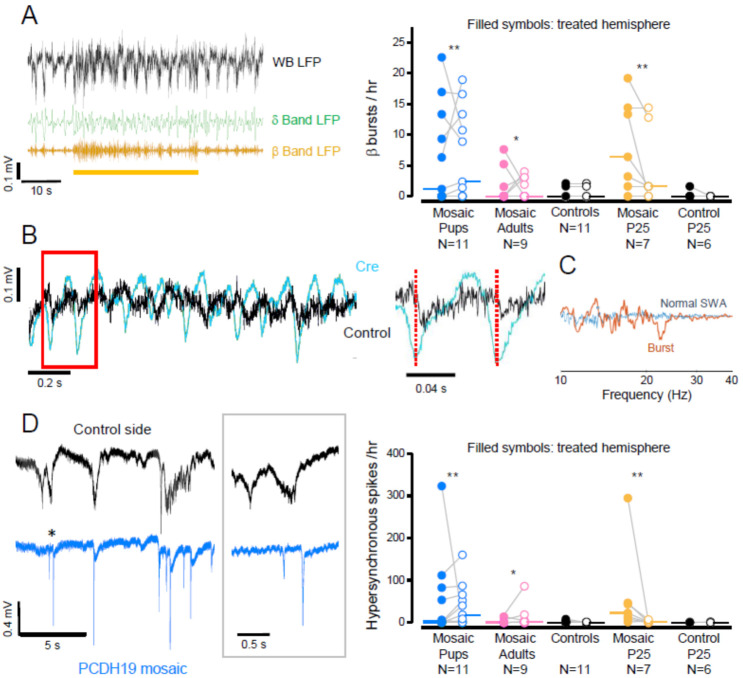
Hyperexcitability of the PCDH19 mosaic. (**A**) Left: example of a burst in the β/σ (9–25 Hz) frequency band of the LFP recorded in the PCDH19 mosaic. The burst (yellow bar) affects the physiological slow waves as shown by the band-passed signals in the δ (0.5–4 Hz, green trace) and β/σ (yellow) bands. Right: Number of β bursts per hour of all recorded animals. In this and following Figures, the dots represent the average value of each animal, and the lines connect corresponding data from the control and injected side (empty and filled symbols, respectively). The medians of each group are represented by horizontal lines (median > 0, ** *p* < 0.002, * *p* < 0.02, Wilcoxon signed rank test). (**B**) The β bursts have a counterpart in the control hemisphere. The red box is magnified on the right panel and shows how each peak of the burst is followed by a corresponding activity peak in the control hemisphere. (**C**) The cross spectrum between the two hemispheres is flat at frequencies larger than 10 Hz during ordinary SWA, whereas during the β bursts it shows multiple peaks, indicating that the two cortices are engrained on similar rhythms. (**D**) Left: The LFP recorded in the injected hemisphere of a mosaic pup (blue trace) shows hypersynchronous spikes while the control hemisphere (black) exhibits relatively normal SWA. The inset shows the first two peaks indicated by the asterisk). Right: Number of hypersynchronous spikes (see Materials and Methods for definition) per hour of all recorded animals (median > 0, ** *p* < 0.002, * *p* < 0.01, Wilcoxon signed rank test).

**Figure 3 cells-11-01939-f003:**
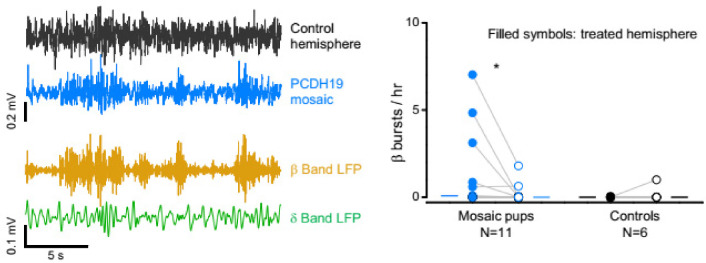
β bursts in the EEG of freely behaving mice. Left: example of oscillations in the β frequency band in EEG recordings. Notice how the oscillations are better defined in the injected hemisphere (blue trace) than in the control hemisphere (black). In this example, the burst in the 9–25 Hz range (yellow) occurs during a period of prominent δ oscillations (green), indicating that the mouse is sleeping (see Supplemental Information for sleep scoring). Right: Number of β bursts per hour. Of the 11 recorded mice, three displayed more than one oscillation per hour (median > 0, * *p* < 0.01, Wilcoxon signed rank test).

**Figure 4 cells-11-01939-f004:**
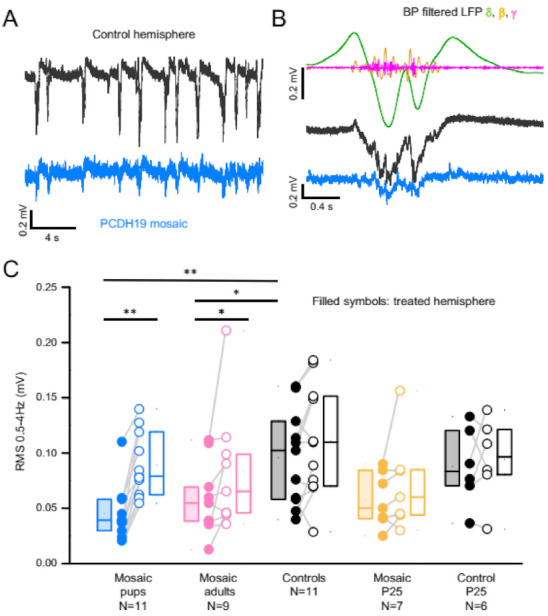
SWA is disrupted in PCDH19 mosaic patches. (**A**) Example LFP of a mosaic pup with reduced US amplitude in the injected hemisphere (blue trace) compared to the control hemisphere (black). (**B**) The traces on the right show a close up of a US, with the band pass filtered signal of the control hemisphere in the δ (0.5–4 Hz; green), β (9–25 Hz; yellow) and γ (40–100 Hz; magenta bands. (**C**) Box plot (median, first and third interquartile) of RMS power in the 0.5–4 Hz range of the LFP. Significant differences are indicated with asterisks (** *p* < 0.005, * *p* < 0.05; Wilcoxon signed-rank test for paired measurements; Mann–Whitney test for non-paired data).

**Figure 5 cells-11-01939-f005:**
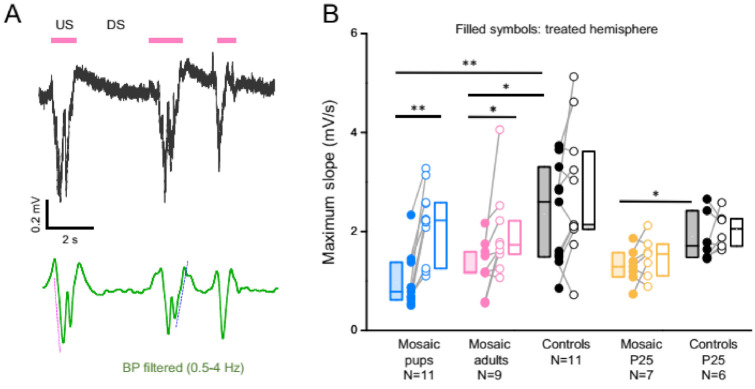
Reduction in SWA is associated with a reduced slope of state transitions. (**A**) Calculation of the US slope. Beginning and end of each US were identified from the LFP trace (black trace; magenta bars indicate the USs). The signal was filtered in the δ band (0.5–4 Hz, green) and we computed the slope of both transitions from DS to US (magenta dashed line) and from US to DS (blue dashed line, see Materials & Methods). (**B**) Box plots (median, first and third interquartile) of the slope of the transition from US to DS. Significant differences are indicated with asterisks (** *p* < 0.005, * *p* < 0.05; Wilcoxon signed-rank test for paired measurements; Mann–Whitney test for non-paired data).

**Figure 6 cells-11-01939-f006:**
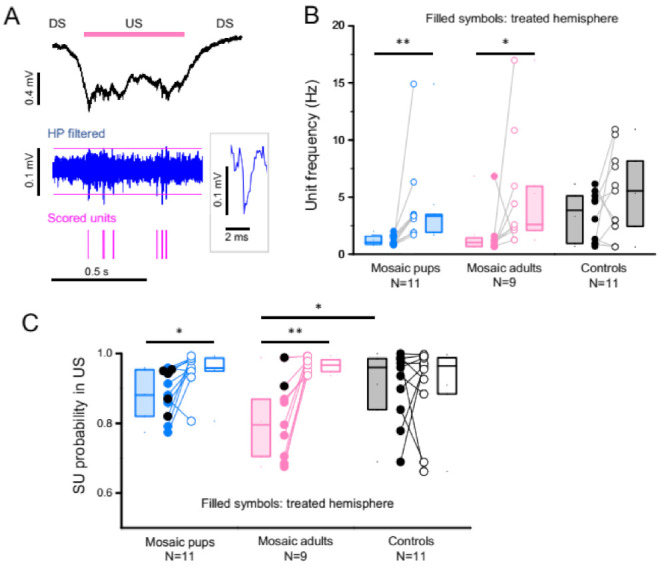
Unit activity is reduced and less synchronized in PCDH19 mosaic tissue. (**A**) Example of an LFP recording of a transition to a US (black trace) and corresponding high pass filtered trace (>300 Hz, blue) of a control animal recorded at P60. The magenta lines indicate the automatically calculated threshold, and the scored units are shown in the lower raster plot in magenta. Note how units cluster in the USs. The inset shows the magnification of a high passed unit. (**B**,**C**) Box plots (median, first and third interquartile) of unit frequency (**B**) and of the probability of units (SU) to occur during Up states (**C**). In panel c, the black symbols represent mice with a hyperexcitable phenotype, defined as more than 5 β bursts per hour. Significant differences are indicated with asterisks (** *p* < 0.005, * *p* < 0.05; Wilcoxon signed rank test for paired measurements; Mann–Whitney test for non-paired data).

**Figure 7 cells-11-01939-f007:**
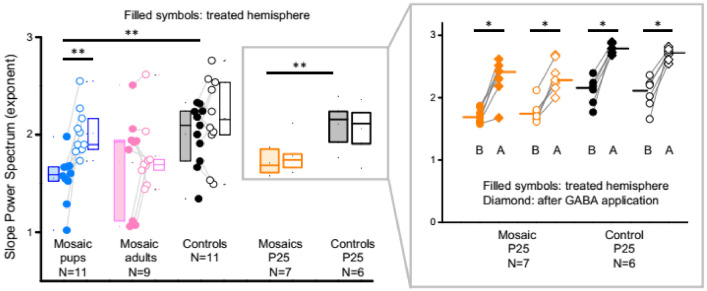
PCDH19 mosaic animals have an increased excitation to inhibition ratio. Box plots of the slope of the LFP power spectrum (median, first and third interquartile). Significant differences are indicated with asterisks (** *p* < 0.005, * *p* < 0.05; Wilcoxon signed rank test for paired measurements; Mann–Whitney test for non-paired data). Power spectrum slopes were significantly reduced in ‘Mosaic pups’ and in animals recorded at P25. The inset contains the values of the same P25 mice shown in the main graph (boxed area) at the same scale and shows the values after ectopic application of GABA (B: before GABA; A: after GABA). Bars show median values, dots show the average value per animal with lines connecting the value of each animal before and after GABA application. GABA significantly increases the slope in all cases.

**Figure 8 cells-11-01939-f008:**
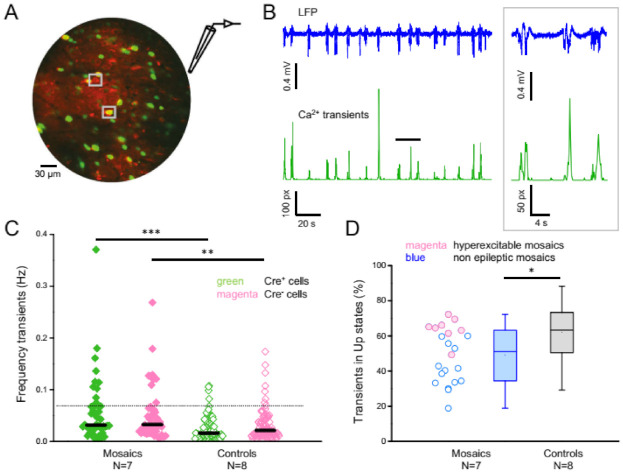
In vivo 2-photon calcium imaging in PCDH19 mosaic brain. (**A**) Example field of view with Cre-negative red neurons (top grey square; expressing jRGECO1a only) and Cre positive (PCDH19 knockout) green neurons (bottom square) that express jRGECO1a in the cytoplasm and EGFP-Cre recombinase in the nucleus. (**B**) Calcium transients (green trace) measured over the entire field of view. The vertical axis displays the number of pixels affected by calcium activity over time (see Supplemental Information). The black trace shows the LFP recorded from the same hemisphere with an electrode positioned just outside of the imaged field at a depth of 250 µm. The inset shows magnified traces in correspondence of the black bar. Notice how Ca^2+^ transients are phase locked to the USs. (**C**) Frequency of calcium transients in red and green cells of mosaic mice and controls. Each dot represents the average value for each active neuron. Significant differences are indicated with asterisks (*** *p* < 0.0005, ** *p* < 0.005; Mann–Whitney test). Transient frequency is significantly increased in mosaics compared to control mice both for Cre+ (PCDH19 KO) neurons and for Cre- neurons. (**D**) Box plots of the percentage of Ca^2+^ transients occurring within USs, demonstrating a significant reduction in synchronization with the SWA oscillation in mosaic mice compared to control mice (* *p* < 0.02; Mann–Whitney test, whiskers indicate data range). Each dot represents the value obtained by all neurons in the imaging field of PCDH19 mosaic mice. The filled magenta dots represent the values obtained from mosaic animals with a hyperexcitable (>5 β oscillations per hour) phenotype, the blue dots from all other mosaics.

## Data Availability

Specific code used in the dt analysis has been deposited on GitHub as specified in the methods. All data will be posted on GitHub following the outcome of the review process.

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
