# Peer review of "Perturbation of Cortical Excitability in a Conditional Model of PCDH19 Disorder"

_cells, 2022, doi:10.3390/cells11121939_

Round 1

Reviewer 1 Report

The Manuscript by Lamers et al. (Manuscript ID: cells-1756727) on " Perturbation of cortical excitability in a conditional model of PCDH19 disorder" aimed to study the effects of mosaic PCDH19 expression on cortical networks. Authors mimicked the pathology of DEE9 by introducing a patch of mosaic protein expression in the cortex of conditional PCDH19 knockout mice and characterized the physiology of the disrupted network. The authors found transient episodes of hyperexcitability and disturbed slow-wave activity. Authors reported that the focal mosaic mutation of PCDH19 disrupts network cortical computation. In my view, the study is promising, and the experiments performed in this study support the conclusion of the manuscript. I have some comments that can help to improve the quality of the manuscript as follows:

-       The abstract needs to be updated, summarizing all key aspects of the manuscript.

-       Authors mentioned that focal mosaic mutation of PCDH19 disrupts network cortical computation. Do the authors think of any other reasons? Are there any other possibilities for disturbed network cortical computation?

-       The manuscript needs to be well polished. The language should be lucid, which will help readers to easily understand the contents.

-       It will be better if the authors give a figure that can represent the overall conclusion of the manuscript. This will help the readers to understand the conclusion easily.

-       What are the possibilities for the use of “Perturbation of cortical excitability” for diagnosis of PCDH19 disorder or PCDH19 epilepsy (DEE9)? This needs to be explained in the discussion section of the manuscript.

-       The list of references is not uniform. Many references have a DOI number, however, some references do not have DOI numbers. Authors are suggested to keep the uniformity in the list of references.

Author Response

We are grateful for the comments and suggestions that we have addressed as follows:

The abstract needs to be updated, summarizing all key aspects of the manuscript.

We have revised the abstract accordingly while respecting the word limit of 200 words.

Authors mentioned that focal mosaic mutation of PCDH19 disrupts network cortical computation. Do the authors think of any other reasons? Are there any other possibilities for disturbed network cortical computation?

The induction of the PCDH19 mosaic requires the injection of a viral vector that could activate an inflammatory response that, in principle, could impinge on cortical development. That is why our control group is formed by mice that received a unilateral AAV injection of either the same virus used for the mosaic induction (in WT mice) or of a virus only carrying the GFP gene in floxed PCDH10 mice. These controls returned similar results and therefore we are convinced that the physiological phenotype observed within the area carrying the mutation is due to the PCDH19 loss.

Having said that, our data show that ectopic activity can be detected also in the control area and this disruption is not due directly to the mutation but to the propagation of the activity burst through long range connection. Examples of this can be found in figure 2 where we show that epileptiform burst in the 9-25 Hz band are also apparent in the controlateral hemisphere. We have added two panels to figure 2 showing in detail the morphology and spectral properties of the bursts in the PCDH19 mosaic and in the control area. 

The manuscript needs to be well polished. The language should be lucid, which will help readers to easily understand the contents.

We have tried to clarify some sections that were very technical and we tried to provide a better background of the experiments. Changes are indicated in track change in the submitted revised manuscript.

It will be better if the authors give a figure that can represent the overall conclusion of the manuscript. This will help the readers to understand the conclusion easily.

We have prepared a summary figure that can be used as a graphic abstract as fom Author Instructions. We hope that this figure adequatley describe the overal ideas at the basis of the study.

What are the possibilities for the use of “Perturbation of cortical excitability” for diagnosis of PCDH19 disorder or PCDH19 epilepsy (DEE9)? This needs to be explained in the discussion section of the manuscript.

This is a very important point that has been basically ignored in the discussion. Indeed, several aspects of the electrophysiological signatures that we have found could potentially be used as biomarkers of the disease and of the efficacy of any pharmacological treatment. This point has been addressed at the end of the Discussion at page 13.

The list of references is not uniform. Many references have a DOI number, however, some references do not have DOI numbers. Authors are suggested to keep the uniformity in the list of references.

References have been checked for uniformity. We are grateful to the Reviewer that went  through the task of checking the references!

Reviewer 2 Report

PCDH19 epilepsy (Developmental and Epileptic Encephalopathy) is an epileptic encephalopathy caused by mutations in the X-linked gene Protocadherin 19  likely to be the second most frequent genetic cause of epilepsy in females, and it is associated with intellectual disability, autism spectrum disorder and late-onset schizophrenia. Research done on mouse models of  PCDH19 epilepsy suggest that PCDH19 regulates cell adhesion and network formation. In this work, using in vivo 60 electrophysiological and imaging studies in a novel mouse model of the disease, the authors asked whether cortical network wiring and computation are affected by mosaic PCDH19 expression. To this end, they created a patch of mosaic PCDH19 expression by injecting an AAV expressing EGFP-Cre recombinase in the right occipital cortex of a conditional mouse where exon 3 of the murine gene is flanked by a pair of LoxP sequences. They report that even if PCDH19 mosaicism occurs only in a relatively small portion of the cortex, these mice show a hyperactive behavior and a complex electrophysiological phenotype, consistent with the clinical features of the disease. 

This is a nice piece of technical work. However, I was surprised that the authors did not discussed at al the potential role of neurogenesis in their model (see, DOI: 10.1016/S0002-9440(10)63897-7)

Author Response

We are glad that the Reviewer found merit in our study and we have addressed the raised point:

However, I was surprised that the authors did not discussed at al the potential role of neurogenesis in their model (see, DOI: 10.1016/S0002-9440(10)63897-7)

It is potentially possible that the very early phases of development, including neurogenesis and migration during cortical layering, might be affected by the loss of PCDH19 as suggested by recent studies from Laura Cancedda's lab. However, in our case, the genetic damage is imposed in post natal neurons that have already completed migration and we are not affecting progenitors of cortical neurons that are not present anymore at the AAV-CRE injections site at this developmental stage.